# Surgical vs. Conservative Management of Patients with Nonfunctioning Pancreatic Neuroendocrine Tumors Smaller than 2 cm (NF-PANNETs < 2 cm) Systematic Review and Meta-Analysis

**DOI:** 10.3390/cancers17101649

**Published:** 2025-05-13

**Authors:** Giuseppe Sena, Giuseppe Currò, Giuseppina Vescio, Giorgio Ammerata, Angela Amaddeo, Antonia Rizzuto

**Affiliations:** 1Department of General Surgery, “Renato Dulbecco” Hospital, Viale Europa, 88100 Catanzaro, Italy; 2Department of Health Science, “Magna Graecia” University, Viale Europa, 88100 Catanzaro, Italy; 3Department of Surgical and Medical Science, “Magna Graecia” University, Viale Europa, 88100 Catanzaro, Italy; vescio@unicz.it (G.V.); angela.amaddeo@gmail.com (A.A.); arizzuto@unicz.it (A.R.); 4Department of General Surgery, “Di Venere” Hospital, Via Ospedale Di Venere, 1, 70131 Bari, Italy; giorgiosala4@gmail.com

**Keywords:** pancreatic neuroendocrine tumors, small nonfunctional pancreatic neuroendocrine tumor, surgical management, conservative management

## Abstract

This meta-analysis and systematic review revealed that surgical intervention for nonfunctioning pancreatic neuroendocrine tumors smaller than 2 cm significantly improves the overall survival rate compared with conservative treatment but does not improve cancer-specific survival. There is a variation in outcomes across different studies. While surgery may be beneficial for some patients, the absence of clear benefits in cancer-specific survival and the potential risks involved highlight the need for personalized decision-making. Therefore, a conservative approach that includes active surveillance may be more appropriate for low-risk patients.

## 1. Introduction

Pancreatic neuroendocrine tumors (PANNETs) are sporadic malignancies with an incidence of 0.43 × 100,000, representing 1% to 2% of all pancreatic tumors [1,2,3,4,5]. Most PANNETs occur sporadically. However, 10% of cases are associated with genetic conditions such as multiple endocrine neoplasia type 1 (MEN1), neurofibromatosis type 1, von Hippel–Lindau disease, and tuberous sclerosis [6].

According to the National Comprehensive Cancer Network (NCCN) guidelines, functional PANNETs and nonfunctional PANNETs (NF-PANNETs) larger than 2 cm should undergo surgical resection [7]. However, no consensus exists on managing nonfunctioning PANNETs smaller than 2 cm. Therefore, their treatment remains controversial [8].

Recently, the World Health Organization has introduced a classification to stratify the risk of malignancy and facilitate treatment planning. The classification considers PANNETs benign with the following characteristics: ≤2 cm, confined to the pancreas, nonangioinvasive, with ≤2 mitosis/10 HPF, and ≤2% Ki-67-positive cells. The major limitation of this classification is that it requires a surgical pathological evaluation, making the preoperative tools of little value [9]. Therefore, when it is impossible to identify tumors with aggressive biology, it is tough to establish the correct treatment for small NF-PANNETs. The guidelines of the European Neuroendocrine Tumor Society (ENETS) establish that, in NF-PANNETs < 2 cm at a low–moderate risk of malignancy, there are no data on the efficacy of surgical treatment [10].

Furthermore, the morbidity associated with pancreatic resections remains exceptionally high. Therefore, the possibility of a surgical cure must be well weighed against the morbidity, mortality, and long-term complications associated with pancreatic surgery [11]. The aim of this study, by literature review and meta-analysis, is to establish the best management of NF-PANNETs < 2 cm based on overall survival (OS) and cancer-specific survival (CSS).

## 2. Materials and Methods

### 2.1. Study Selection

We performed a systematic review and subsequent meta-analysis of all prospective and retrospective studies comparing the OS and CSS of patients with NF-PANNETs < 2 cm undergoing surgical management (SM) or conservative management (CM). OS refers to survival affected by any cause of death, while CSS refers to survival affected specifically by tumor progression-related causes. The systematic review followed the recommendations of the Preferred Reporting Items for Systematic Reviews and Meta-Analyses (PRISMA) [12]. The protocol has not been registered. An extensive online search was conducted using the MEDLINE, EMBASE, Google Scholar, Scopus, Web of Science, and Cochrane Central databases. All articles published up to May 2023 were extracted using the following MeSH terms in all combinations: “pancreatic neuroendocrine tumor”, “small nonfunctional pancreatic neuroendocrine tumor”, “surgical management”, “conservative management”, and “overall survival”. The “related articles” feature on PubMed was utilized to improve the search process, and the references of each potentially eligible article were reviewed. Only English-language articles were included, irrespective of ethnicity or geographical origin. A manual search was conducted to minimize the risk of bias. The two authors who reviewed the papers made the final decision on eligibility.

### 2.2. Inclusion and Exclusion Criteria

In our analysis, we included the following criteria for selecting studies: (1) all retrospective and prospective studies that compare the outcomes of SM versus CM in patients with NF-PANNETs < 2 cm; (2) studies that report overall survival for a minimum of five years; and (3) studies that adequately report overall survival suitable for pooled analysis. We excluded studies that did not meet these criteria and the following criteria: (1) non-comparative studies, (2) case series, (3) animal studies, and (4) studies involving patients with genetic syndromes such as MEN 1.

### 2.3. Data Extraction and Quality Assessment

Two authors independently extracted data, and a third performed a final check. A collegial meeting was necessary to resolve any disagreements. All the data were transferred to a collection database. The Population, Intervention, Control, Outcome(s) (PICO) search framework was used for the search and data extraction. The content of this framework is defined as follows:Population: patients with NF-PANNETs < 2 cm.Intervention: all curative surgical resections.Control: patients undergoing conservative management.Outcome: 5 years OS, 5 years CSS.

For each study, the following characteristics were reported: year of publication, first author, study design, country, and number of patients. The characteristics of the population were also extracted: age, sex, number of patients in the SM and CM groups, tumor size, and histological grade. The outcomes of interest considered for SM versus CM were OS and CSS. Quality assessment was performed using the Risk Of Bias In Non-randomized Studies of Interventions (ROBINS-I) tool [13]. The ROBINS-I tool evaluates bias risk in non-randomized studies comparing intervention effectiveness between two patient groups. It assesses factors such as confounding bias, selection bias, classification bias, deviations from intended interventions, missing data, measurement bias, and reporting bias. Each factor is rated as “low risk”, “moderate risk”, “serious risk”, “critical risk”, or “no information”.

### 2.4. Statistical Analysis

Continuous variables were reported as means and standard deviation, and dichotomous variables were reported as percentages. The Chi-Square and Fisher exact test analyzed dichotomous variables by calculating the odds ratio (OR) and corresponding 95% confidence interval (CI). All *p* values < 0.05 were considered statistically significant. The fixed-effect model and the random-effect model (weighted by the Mantel–Haenszel or DerSimonian and Laird method, respectively) were used for the pooled analyses. Heterogeneity between the studies was quantified using the Chi-Square (or Cochran’s Q statistic) and I^2^ statistic [14,15]. Homogeneity was absent if the Q statistic showed *p* < 0.20, and the heterogeneity was considered significant when the I^2^ statistic exceeded 50%. A meta-regression analysis was conducted to identify sources of heterogeneity, specifically focusing on mean age, tumor location (head or body/tail), and histological grade (well differentiated or moderately differentiated). Additionally, a subgroup analysis was performed based on the study design (monocentric or multicentric) and the region of origin. All the statistical analyses were performed using R version 4.4.3 (R Foundation for Statistical Computing, Vienna, Austria).

## 3. Results

### 3.1. Literature Search

The initial search identified 295 records. Subsequent deduplication and evaluation of titles/abstracts identified 43 studies for complete text analysis. Of these, 38 were excluded because they did not meet the inclusion criteria. Therefore, only six studies were included in the quantitative analysis [8,16,17,18,19,20]. The screening process is summarized in Figure 1.

### 3.2. Characteristics of the Studies

The studies were from Western and Eastern countries (three from the USA, one from England, and two from China) and were published between 2014 and 2021. All the studies had a retrospective design. In three studies, the tumor site was reported and, in three, the histological grade. Three studies did not report CSS and, therefore, could not be included in the pooled analysis for CSS. One was a single-center study, and five were multicenter. The observation period ranged from 1993 to 2015. The characteristics of the studies are summarized in Table 1.

According to the ROBIN-I checklist, all the studies were assessed to have a moderate risk of bias. The risk of bias is summarized in Table 2.

### 3.3. Characteristics of the Patients

The overall number of patients with NF-PANNETs < 2 cm was 4713, of which, 2968 were managed operatively and 1109 were managed conservatively. In all the studies, the patients underwent major resections (pancreaticoduodenectomy, distal pancreatectomy, central pancreatectomy) and enucleations. Positive margins after surgery were reported in two studies.

The mean age of the patients ranged from 56 to 59 years in the SM group and from 62 to 63 years in the CM group. The percentage of women ranged from 41.5% to 54.9% in the SM group and from 51.9% to 59.6% in the CM group. Most lesions were located on the body/tail (59.3–67.6%) rather than on the head (28.4–37.4%). The frequency of the G1/G2 histological grade was higher than that of G3 in both groups. The characteristics of the patients are summarized in Table 3.

### 3.4. Surgery Versus Nonsurgical Management

A pooled analysis of all the data demonstrated increased OS in patients managed operatively compared with those managed conservatively at five years (OR = 1.77, 95% CI: 0.96 to 2.58; *p* = 0.002). According to the Q-test and the I^2^ test, the outcomes appeared to be heterogeneous (Q = 43.98, *p* < 0.001, tau^2^ = 0.46, I^2^ = 88.63%) (Figure 2). In contrast, the meta-analysis did not demonstrate increased CSS in patients undergoing surgical resection compared with conservative management (OR = 1.01, 95% CI: −5.25 to 7.27; *p* = 0.56). In this case also, the outcomes appeared to be heterogeneous (Q = 22.81, *p* < 0.0001, tau^2^ = 1.72, I^2^ = 91.23%) (Figure 3). In the meta-regression analysis, the mean age (*p* = 0.153), tumor location (head *p* = 0.154; body/tail *p* = 0.830), and histological grade (well differentiated *p* = 0.742; moderately differentiated *p* = 0.953) were not found to be significant moderators (Table 4). The subgroup analysis did not show a statistically significant increase in OS for the SM group compared with the CM group, with studies from Western countries reporting an OR of 1.39 (95% CI: −0.13 to 2.90; *p* = 0.062) (Figure 4) and studies from Eastern countries showing an OR of 2.27 (95% CI: −3.24 to 7.77; *p* = 0.120) (Figure 5). Conversely, the analysis of multicenter studies indicated a statistically significant increase in OS for the SM group compared with the CM group, with an OR of 2.05 (95% CI: 1.33 to 2.76; *p* = 0.001) (Figure 6).

## 4. Discussion

Most published data have shown that NF-PANNETs > 2 cm are associated with poor prognostic factors such as lymph node invasion, distant metastasis, and local invasion. However, there are limited data regarding tumors smaller than 2 cm on these prognostic factors. In particular, the only nonsurgical prognostic factors that can be evaluated are size and functional status. Based on the suspected tumor type, laboratory tests commonly assess hormone levels such as insulin, gastrin, glucagon, vasoactive intestinal peptide (VIP), and somatostatin to identify functioning PANNETs.

These patients should undergo a biopsy or surgery to obtain tumor tissue and lymph node samples for histological evaluation [21]. Surgical resection is the typical treatment for NF-PANNETs, but it is challenging for nonfunctioning tumors < 2 cm, as demonstrable survival benefits are lacking. Therefore, as we specified above, the ENETS and NCCN suggest cautious observation as appropriate management for NF-PANNETs < 2 cm without dilation of the main pancreatic duct, particularly for pancreatic head tumors, for patients with various comorbidities, and for patients judged to be at a high surgical risk [10,22].

Before 2010, the World Health Organization (WHO) classified pancreatic neuroendocrine tumors (PANNETs) using a hybrid classification system that combined stage and grade information into a single prognostic prediction [23]. The WHO categorizes tumors based on their morphology and proliferative activity, which includes the number of mitoses observed per high-resolution field and the Ki-67 index. This information can be obtained through endoscopic ultrasound biopsy or CT/US-guided transabdominal biopsy [23].

Additionally, the WHO recommends monitoring nonfunctional PANNETs smaller than 2 cm due to their low growth rate, minimal incidence of lymph node metastasis, and the significant risks associated with the surgery required to remove them. Imaging follow-up for non-resected patients according to the RECIST criteria should be conducted every 3 to 6 months for the first 2 years. If the tumor remains stable after this period, follow-up imaging should occur every 6 to 12 months. This should include a combination of contrast-enhanced CT scans, whole-abdominal MRI, and SST-PET/CT every 1 to 2 years if clinically indicated. Several studies support this approach. A retrospective analysis of 145 patients found that only 3.9% of nonfunctional PANNETs smaller than 2 cm were malignant. It is, therefore, understandable how the risks and benefits of surgical resections must be weighed in patients with small lesions, given the potential mortality and overall complication rate; however, patients over 55 years of age more frequently had high-grade tumors [24].

In a report of 133 patients with NF-PANNETs < 4 cm, Lee et al. demonstrated that the group of patients treated conservatively had no disease progression or cancer-specific mortality after 45 months of follow-up [25]. In another multicenter study on 46 patients with NF-PANNETs < 2 cm, the absence of nodal or distant metastases was demonstrated after a mean follow-up of 34 months [26]. Furthermore, a recent nationwide cohort study of 76 patients with NF-PANNETs < 2 cm found that watchful waiting is a safe alternative to upfront surgery. It also revealed poor quality of life for patients undergoing surgery [27]. However, in several reports, NF-PANNETs < 2 cm have shown signs of malignancy, such as extrapancreatic extension, lymph node metastasis, distant metastasis, and recurrence, leading to an increase in cancer-related death [28]. In particular, Lombardi et al. reported that NF-PANNETs < 2 cm were grade II or greater in 26% of cases, stage II or more in 30.4% of cases, and 17.4% had distant metastases.

Furthermore, in this study, exclusion from surgery represented undertreatment for 39% of patients [29]. In their study involving 139 patients who underwent resection for any asymptomatic PANNETs, Haynes et al. showed that 8% of 39 patients with tumors smaller than 2 cm developed recurrence and died of the disease [30]. The Chinese Study Group for Neuroendocrine Tumors (CSNET) consensus established that a more aggressive approach could be suggested except in selected cases of NF-PANNETs < 1 cm and patients with high surgical risk; it recommended that patients with NF-PANNETs < 2 cm should undergo surgical resection and subsequent careful follow-up [31]. In addition, according to the North American Neuroendocrine Tumor Society (NANETS) guidelines, observation could be a good choice for NF-PANNETs < 1 cm, but the management of NF-PANNETs 1–2 cm should be personalized based on various factors such as age, endoscopic ultrasonography-end-needle aspiration or endoscopic ultrasonography–biopsy findings (grade, Ki-67), comorbidities, anatomical location, tumor growth status, patient preferences, extent of procedure required for complete resection, and access to long-term follow-up [32]. A recent meta-analysis involving 714 patients with NF-PANNETs < 2 cm found that surgical resection was associated with increased OS at 1, 3, and 5 years. Although surgical resection appears to improve survival, only two studies analyzed NF-PANNETs < 2 cm in this work, with a limited sample size and poor statistical significance [33]. Finally, the preliminary results of the ASPEN trial at one year of follow-up demonstrated that active surveillance is the preferred approach for small NF-PANNETs. However, personalized management was necessary for more extensive lesions, young patients, and cases with measurable growth. Surgery was always required for small NF-PANNETs with a dilated mean pancreatic duct [34]. Alternative treatments for small NF-PANNETs are being explored, with preliminary studies suggesting that radiofrequency ablation (RFA) may be effective. However, more data are needed to support its routine use. A prospective study is underway to assess the safety and efficacy of endoscopic ultrasound (EUS)-guided RFA for PANNETs.

Additionally, the current evidence does not support using somatostatin analogs to manage disease progression during the surveillance of small NF-PANNETs, and the patients treated experienced complications related to cholelithiasis, including acute cholecystitis, gangrenous cholecystitis, or intestinal obstruction that necessitated emergency intervention [35]. After all, based on solid biological reasoning, mTOR inhibitors like everolimus significantly improved the progression-free survival (PFS) compared with placebo in patients with progressive G1 and G2 PANNETs [36]. This meta-analysis investigates the outcomes associated with the surgical versus conservative management of NF-PANNETs < 2 cm, focusing on OS and CSS. The findings provide critical insights into the management of these tumors, which often present a clinical challenge due to their typically indolent nature and the potential risks inherent in surgical interventions. The pooled analysis indicates that SM is associated with significantly improved OS at the five-year mark compared with CM. This result suggests a potential survival advantage for patients who undergo surgical resection. However, significant heterogeneity was noted, implying that study design, patient demographics, and tumor characteristics may have influenced the findings.

Subgroup analyses highlighted variability in the outcomes; multicenter studies demonstrated a statistically significant advantage for SM, whereas studies conducted in Western and Eastern regions did not show statistically significant differences. Furthermore, the analysis did not reveal a significant difference in CSS between SM and CM, suggesting that surgery does not confer a distinct benefit in reducing cancer-specific mortality for NF-PANNETs < 2 cm. Similar to the findings on OS, the considerable heterogeneity observed in the CSS analysis cautions against drawing overly broad conclusions. The meta-regression analysis revealed that factors such as mean age, tumor location, and histological grade were not significant moderators of outcomes. This finding may reflect the relatively uniform clinical behavior of NF-PANNETs < 2 cm across these variables. Additionally, subgroup analyses based on geographic regions failed to demonstrate statistically significant differences in OS.

In contrast, multicenter studies consistently reported improved OS for SM, likely indicating more standardized surgical practices and patient selection criteria. These results underscore the complexities involved in the decision-making process for managing small NF-PANNETs. While surgical management appears to provide a survival advantage regarding OS, the lack of a corresponding improvement in CSS implies that the observed benefits may not be solely attributable to tumor-related factors. Instead, enhanced OS may reflect broader variations in patient care, comorbidities, or other unmeasured confounders between the SM and CM groups. It is essential to consider the risks associated with pancreatic surgery, including morbidity and potential impacts on quality of life, in the context of the modest survival benefits reported. Notably, patients with well-differentiated, indolent tumors may experience minimal additional benefit from surgical intervention, thereby favoring a conservative management approach involving active surveillance. This study represents one of the most comprehensive meta-analyses comparing surgical and conservative management in this patient population. Including multicenter studies and data from different geographical regions enhances the generalizability of the findings. However, several limitations warrant attention:Heterogeneity: Significant variability in OS and CSS analyses limits the reliability of pooled estimates. Although meta-regression and subgroup analyses have aimed to address this issue, other confounding factors are likely unconsidered.Study Design: The predominance of retrospective studies introduces potential biases, including selection bias and variations in baseline characteristics between groups.Limited Long-term Data: The analysis primarily focused on five-year outcomes, indicating the need for additional long-term survival data to better understand these tumors’ natural history.

Future prospective multicenter studies are warranted to refine the patient selection criteria and establish more precise management guidelines for NF-PANNETs < 2 cm. While randomized controlled trials present challenges in this context, they could help address residual confounding factors and provide more definitive evidence regarding the comparative benefits of surgical versus conservative management. Additionally, incorporating molecular and genetic profiling into clinical decision-making has the potential to facilitate more personalized management approaches.

## 5. Conclusions

The present systematic literature review and meta-analysis demonstrated that the surgical management of NF-PANNETs < 2 cm is associated with improved OS but does not significantly benefit CSS. There is considerable variability in outcomes across different studies. While surgery may benefit certain patients, the absence of a clear advantage in CSS and the potential risks associated with surgery highlight the necessity of individualized, multidisciplinary decision-making. Therefore, a conservative approach involving active surveillance may still be the most appropriate option for low-risk patients.

## Figures and Tables

**Figure 1 cancers-17-01649-f001:**
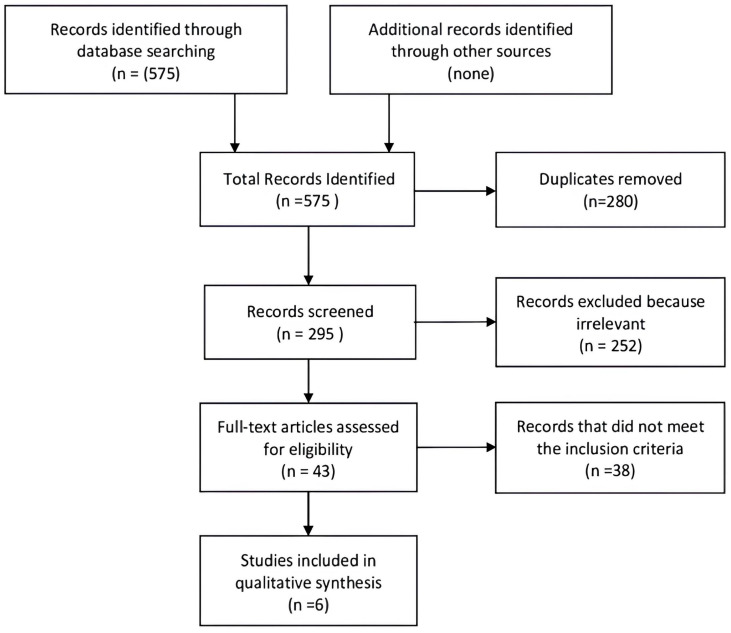
Prisma flow chart of the selection process.

**Figure 2 cancers-17-01649-f002:**
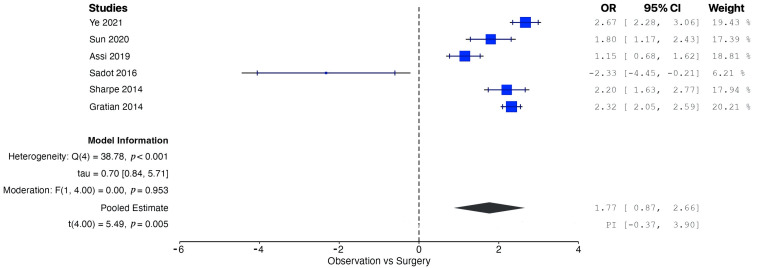
Forest plot for pooled analysis of the overall survival [8,16,17,18,19,20].

**Figure 3 cancers-17-01649-f003:**
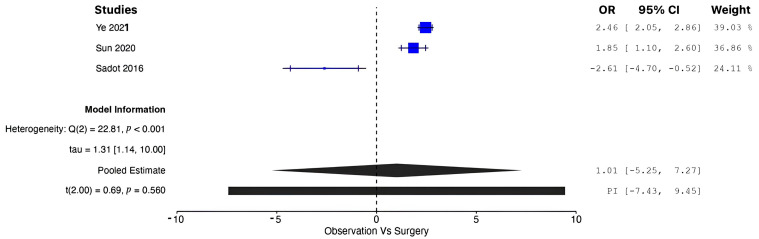
Forest plot for pooled analysis of cancer-specific survival [16,19,20].

**Figure 4 cancers-17-01649-f004:**
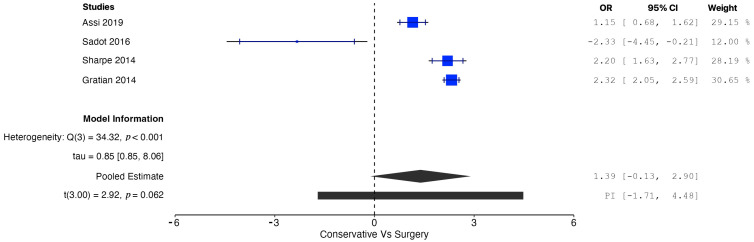
Forest plot for pooled analysis of the overall survival of studies from Western countries [8,16,17,18].

**Figure 5 cancers-17-01649-f005:**
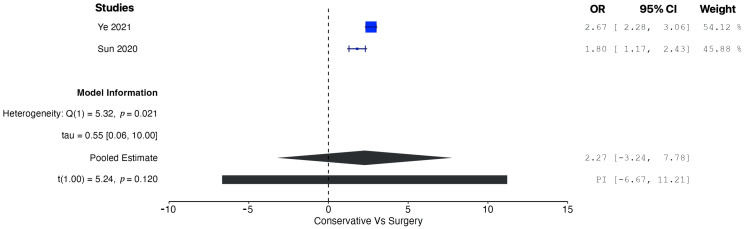
Forest plot for pooled analysis of the overall survival of studies from Eastern countries [19,20].

**Figure 6 cancers-17-01649-f006:**
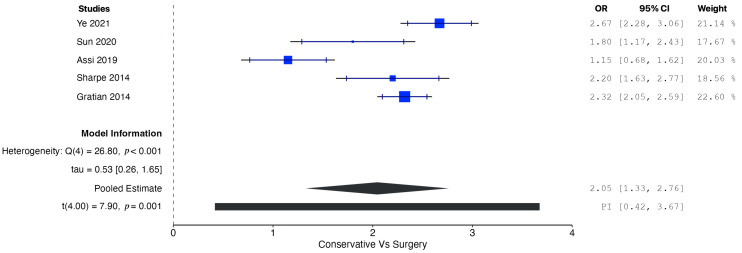
Forest plot for pooled analysis of the overall survival of multicenter studies [8,16,18,19,20].

**Table 1 cancers-17-01649-t001:** Characteristics of the studies.

References	Country	Year	Study Design	Number of Patients
Gratian [16]	England	2014	Retrospective Monocentric	1367
Sharpe [17]	USA	2014	Retrospective Monocentric	380
Sadot [8]	USA	2016	Retrospective Multicentric	187
Assi [18]	USA	2019	Retrospective Multicentric	1014
Sun [19]	China	2020	Retrospective Multicentric	759
Ye [20]	China	2021	Retrospective Multicentric	1006

**Table 2 cancers-17-01649-t002:** Risk of bias.

References	Baseline Confounding	Selection of Participants	Classification of Intervention	Deviation from Intended Intervention	Missing Data	Measurement of Outcomes	Selection of Reported Results	Overall Risk of Bias
Gratian [16]	Low	Low	Low	Moderate	Low	Moderate	Low	Moderate
Sadot [17]	Low	Low	Moderate	Low	Low	Moderate	Low	Moderate
Sharpe [8]	Low	Moderate	Low	Low	Low	Low	Low	Moderate
Assi [18]	Moderate	Moderate	Moderate	Low	Moderate	Moderate		
Sun [19]	Moderate	Moderate	Low	Low	Moderate	Low	Low	Moderate
Ye [20]	Moderate	Low	Moderate	Low	Low	Moderate	Low	Moderate

**Table 3 cancers-17-01649-t003:** Characteristics of the patients.

References	Number of Patients (%)		Age (Y)		Female (%)		Tumor Site (%)		Histological Grade (%)	
	SM	CM	SM	CM	SM	CM	Head	Body/Tail	SM	CM
Gratian [16]	999 (73.07)	368 (26.92)	56	63	556 (55.7)	205 (55.7)	N/A	N/A	G1:859 (85.9), G2:95 (9.5), G3:31 (3.1)	G1:191 (52.0), G2:58 (15.7), G3:110 (29.9)
Sadot [17]	77 (41.1)	104 (55.6)	59	63	32 (41.5)	54 (51.9)	70 (37.4)	111 (59.3)	N/A	N/A
Sharpe [8]	71 (18.6)	309 (81.3)	57	62	39 (54.9)	179 (57.9)	123 (32.3)	257 (67.6)	G1:13 (18.3), G2:4 (5.6), G3:10 (14)	G1:116 (37.5), G2:28 (9), G3:20 (6.4)
Assi [18]	890 (87.7)	124 (12.22)	N/A	N/A	N/A	N/A	N/A	N/A	N/A	
Sun [19]	706 (93)	53 (6.9)	N/A	N/A	N/A	N/A	N/A	N/A	N/A	N/A
Ye [20]	855 (84.9)	151 (15)	N/A	N/A	451 (52.7)	90 (59.6)	286 (28.4)	608 (60.4)	G1/G2:834 (97.5), G3:21 (13.9)	G1/G2:128 (84.7), G3:23 (15.2)

N/A: Not Available.

**Table 4 cancers-17-01649-t004:** Effect size meta-regression term tests.

	F	df_1_	df_2_	*p*
Mean Age	3.096	1	4.000	0.153
Head	0.093	1	4.000	0.776
Body/Tail	0.052	1	4.000	0.830
Well Differentiated	0.124	1	4.000	0.742
Moderate Differentiated	0.004	1	4.000	0.953

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
