# Peer review of "Surgical vs. Conservative Management of Patients with Nonfunctioning Pancreatic Neuroendocrine Tumors Smaller than 2 cm (NF-PANNETs < 2 cm) Systematic Review and Meta-Analysis"

_cancers, 2025, doi:10.3390/cancers17101649_

Round 1
Reviewer 1 Report
Comments and Suggestions for Authors
In this systematic review and meta-analysis, the authors evaluated whether surgical management is better in terms of overall survival and cancer-specific survival compared with conservative management of non-functioning pancreatic neuroendocrine tumors smaller than 2 cm (nf-pan-NETs <2cm)
A total of six studies were found to fulfill the inclusion and exclusion criteria and these were included in the meta-analysis.
It was shown that surgical management for nf-pan-NETs <2cm improves 5-year overall survival significantly compared to conservative treatment. However, there were no differences regarding cancer-specific survival.
Overall the methodology and the analysis are proper and well conducted. However the inclusion of only six studies and their retrospective nature introduce potential biases such as selection bias and variations in baseline characteristics between the two study groups. Nevertheless, the authors present a comprehensive discussion and the potential limitations of the study are well addressed.
A minor comment: Legends to the figures should be added
Author Response
We thank the reviewer for the words of appreciation on our work.
We have inserted the legends to the images as a suggestion
Reviewer 2 Report
Comments and Suggestions for Authors
This is a very interesting study on the most appropriate management of non-functioning pancreatic neuroendocrine tumors smaller than 2 cm.
The study is already very extensive, but to make it easier to read, the following terms should be more clearly defined:
1) Difference between "overall survival" and "cancer-specific survival"
2) What laboratory tests would be necessary to determine whether a pancreatic neuroendocrine tumor is non-functioning?
3) Whether endoscopic ultrasound biopsy would be sufficient to characterize the number of mitoses and the percentage of Ki-67-positive cells.
4) What follow-up regimen is proposed: endoscopic ultrasound, laboratory tests, magnetic resonance imaging, etc.
Line 231 the last words look like a mistake.
Author Response
We are very grateful to the reviewer for his valuable suggestions:
- We wrote a statement about it in the methods session.
- We elaborated on the point in the discussion session.
- We wrote about the possibility of doing a transabdominal biopsy in the discussion session.
- 4 We wrote about the follow-up in the discussion session.
Reviewer 3 Report
Comments and Suggestions for Authors
The present study aimed to establish the best management of NF-PANETs <2cm based on overall survival (OS) and cancer-specific survival (CSS) by literature review and meta-analysis. The results showed that surgical management of NF-PANETs < 2 cm improves overall survival (OS) but does not significantly enhance cancer-specific survival (CSS). There is variability in outcomes among studies, and while surgery may help some patients, the lack of clear CSS benefits and associated risks call for individualized decision-making. Therefore, a conservative approach with active surveillance may be more suitable for low-risk patients. The study was overall well designed and written. Some minor points are listed as below.
1. The significant heterogeneity was observed in both OS and CSS analyses. The authors should try to explore potential sources of the heterogeneity.
2. All included studies are retrospective, introducing inherent selection bias. Therefore, limitations should be added concerning this point.
Author Response
We thank the reviewer for the interesting suggestions.
1 We have attempted to explain the heterogeneity emerged in the pooled analysis by metaregression and subgroup analysis and we have mentioned it in the discussion session
2.We have written a statement regarding the presence of retrospective studies as potential factors of bias in the discussion session
Reviewer 4 Report
Comments and Suggestions for Authors
Comments on non-functioning neuroendocrine tumor less than 2 cm
As the authors mentioned, there is no consensus for the best selection for management of non-functioning neuroendocrine tumor of the pancreas that measured less than 2 cm. After searching for answers in the literature, the results still lack definite evidence to establish reliable criteria and actually make the readers feel confused.
The content of this manuscript has several defects and needs revision before consideration for acceptance for publication, as follows:
- There are 5 articles that reviewed by the authors. In them, can the authors have persuasive explanation about the reason why in references 8, 16 and 19, there are significant number of patients that the histology is grade 3, but the researchers left them to receive only conservative management? If grade 3 means ‘mitosis is fast and carry a high potential tendency of malignancy’ is correct.
- The authors stated that「 ..in subgroup analysis did not show a statistically significant increase in OS ..」. In this article, there is no clear definition about what is ‘subgroup’?
- The authors search the database and find 5 articles for systemic review. However, in the discussion section, the authors cited several references, from 25 to 34 and said something related to the topic, why these papers did not included in the review pool?
- For getting the readers away from confusion, I suggested that the authors should review the articles more carefully. The author might elicit some useful information from so many papers so as to tell the readers some guidelines for management of non-functioning neuroendocrine tumors of the pancreas less than 2 cm that having some special character, such as advanced grading or dilated pancreatic duct.
Author Response
We thank the reviewer very much for the comments, to which we respond point by point
- The articles reviewed are 6 and not 5. It is correct that G3 is an aggressive and potentially malignant lesion and therefore should be treated surgically. However, within the cited works, the authors do not explain why these patients were treated conservatively. These were probably patients with comorbidities and not suitable for surgery
Reference 25 PAN-NET’s smaller than 2 cm with a follow-up of less than 5 years, and therefore not meeting the inclusion criteria
- In line 134, what is meant by subgroups is defined
- The works cited by the reviewer were not included in the pooled analysis for the following reasons:
Reference 25: PAN-NET’s smaller than 2 cm with a follow-up of less than 5 years and therefore not meeting the inclusion criteria.
Reference 26: Article that evaluates the natural history of PAN-NET’s for 18 months in patients treated conservatively.
Reference 27: article that prospectively evaluates the natural history of PAN-NET’s in patients treated conservatively.
Reference 28: Work that evaluates PAN-NET’s treated only surgically, without comparison with tumors treated conservatively.
Reference 29: Work that retrospectively evaluates PAN-NET’s treated only surgically.
Reference 30: Work that retrospectively evaluates PAN-NET’s treated surgically.
Reference 31: These are guidelines.
Reference 32: These are guidelines.
Reference 33: This is a meta-analysis in which only two works consider comparing surgery and observation in nf-PAN-NET’s <2cm.
Reference 34: Work showing the preliminary results of an ongoing trial with a 25-month follow-up and therefore not included in the inclusion criteria of our review
Round 2
Reviewer 4 Report
Comments and Suggestions for Authors
I accepted the explanation of the authors, no further comments.